# Model-Based Reinforcement Learning with Adversarial Training for Online Recommendation

**Xueying Bai**[*][‡]**, Jian Guan**[*][§]**, Hongning Wang**[†]
[‡]Department of Computer Science, Stony Brook University
[§] Department of Computer Science and Technology, Tsinghua University
[†] Department of Computer Science, University of Virginia
`xubai@cs.stonybrook.edu, j-guan19@mails.tsinghua.edu.cn`
`hw5x@virginia.edu`

## Abstract

Reinforcement learning is well suited for optimizing policies of recommender systems. Current solutions mostly focus on model-free approaches, which require frequent interactions with the real environment, and thus are expensive in model learning. Offline evaluation methods, such as importance sampling, can alleviate such limitations, but usually request a large amount of logged data and do not work well when the action space is large. In this work, we propose a model-based reinforcement learning solution which models user-agent interaction for offline policy learning via a generative adversarial network. To reduce bias in the learned model and policy, we use a discriminator to evaluate the quality of generated data and scale the generated rewards. Our theoretical analysis and empirical evaluations demonstrate the effectiveness of our solution in learning policies from the offline and generated data.

## 1 Introduction

Recommender systems have been successful in connecting users with their most interested content in a variety of application domains. However, because of users' diverse interest and behavior patterns, only a small fraction of items are presented to each user, with even less feedback recorded. This gives relatively little information on user-system interactions for such a large state and action space [2], and thus brings considerable challenges to construct a useful recommendation policy based on historical interactions. It is important to develop solutions to learn users' preferences from sparse user feedback such as clicks and purchases [11, 13] to further improve the utility of recommender systems.

Users' interests can be short-term or long-term and reflected by different types of feedback [35]. For example, clicks are generally considered as short-term feedback which reflects users' immediate interests during the interaction, while purchase reveals users' long-term interests which usually happen after several clicks. Considering both users' short-term and long-term interests, we frame the recommender system as a reinforcement learning (RL) agent, which aims to maximize users' overall long-term satisfaction without sacrificing the recommendations' short-term utility [28].

Classical model-free RL methods require collecting large quantities of data by interacting with the environment, e.g., a population of users. Therefore, without interacting with real users, a recommender cannot easily probe for reward in previously unexplored regions in the state and action space. However, it is prohibitively expensive for a recommender to interact with users for reward and model updates, because bad recommendations (e.g., for exploration) hurt user satisfaction and increase the risk of user drop out. In this case, it is preferred for a recommender to learn a policy by fully utilizing the logged data that is acquired from other policies (e.g., previously deployed systems)

---

[*]Both authors contributed equally.

instead of direct interactions with users. For this purpose, we take a model-based learning approach in this work, in which we estimate a model of user behavior from the offline data and use it to interact with our learning agent to obtain an improved policy simultaneously.

Model-based RL has a strong advantage of being sample efficient and helping reduce noise in offline data. However, such an advantage can easily diminish due to the inherent bias in its model approximation of the real environment. Moreover, dramatic changes in subsequent policy updates impose the risk of decreased user satisfaction, i.e., inconsistent recommendations across model updates. To address these issues, we introduce adversarial training into a recommender's policy learning from offline data. The discriminator is trained to differentiate simulated interaction trajectories from real ones so as to debias the user behavior model and improve policy learning. To the best of our knowledge, this is the first work to explore adversarial training over a model-based RL framework for recommendation. We theoretically and empirically demonstrate the value of our proposed solution in policy evaluation. Together, the main contributions of our work are as follows:

- To avoid the high interaction cost, we propose a unified solution to more effectively utilize the logged offline data with model-based RL algorithms, integrated via adversarial training. It enables robust recommendation policy learning.

- The proposed model is verified through theoretical analysis and extensive empirical evaluations. Experiment results demonstrate our solution's better sample efficiency over the state-of-the-art baselines [2]

## 2   Related Work

**Deep RL for recommendation** There have been studies utilizing deep RL solutions in news, music and video recommendations [17, 15, 38]. However, most of the existing solutions are model-free methods and thus do not explicitly model the agent-user interactions. In these methods, value-based approaches, such as deep Q-learning [20], present unique advantages such as seamless off-policy learning, but are prone to instability with function approximation [30, 19]. And the policy's convergence in these algorithms is not well-studied. In contrast, policy-based methods such as policy gradient [14] remain stable but suffer from data bias without real-time interactive control due to learning and infrastructure constraints. Oftentimes, importance sampling [22] is adopted to address the bias but instead results in huge variance [2]. In this work, we rely on a policy gradient based RL approach, in particular, REINFORCE [34]; but we simultaneously estimate a user behavior model to provide a reliable environment estimate so as to update our agent on policy.

**Model-based RL** Model-based RL algorithms incorporate a model of the environment to predict rewards for unseen state-action pairs. It is known in general to outperform model-free solutions in terms of sample complexity [7], and has been applied successfully to control robotic systems both in simulation and real world [5, 18, 21, 6]. Furthermore, Dyna-Q [29, 24] integrates model-free and model-based RL to generate samples for learning in addition to the real interaction data. Gu et al. [10] extended these ideas to neural network models, and Peng et al. [24] further apply the method on task-completion dialogue policy learning. However, the most efficient model-based algorithms have used relatively simple function approximations, which actually have difficulties in high-dimensional space with nonlinear dynamics and thus lead to huge approximation bias.

**Offline evaluation** The problems of off-policy learning [22, 25, 26] and offline policy evaluation are generally pervasive and challenging in RL, and in recommender systems in particular. As a policy evolves, so does the distribution under which the expectation of gradient is computed. Especially in the scenario of recommender systems, where item catalogues and user behavior change rapidly, substantial policy changes are required; and therefore it is not feasible to take the classic approaches [27, 1] to constrain the policy updates before new data is collected under an updated policy. Multiple off-policy estimators leveraging inverse-propensity scores, capped inverse-propensity scores and various variance control measures have been developed [33, 32, 31, 8] for this purpose.

**RL with adversarial training** Yu et al. [36] propose SeqGAN to extend GANs with an RL-like generator for the sequence generation problem, where the reward signal is provided by the discriminator at the end of each episode via a Monte Carlo sampling approach. The generator takes sequential actions and learns the policy using estimated cumulative rewards. In our solution, the generator consists of two components, i.e., our recommendation agent and the user behavior model, and we

model the interactive process via adversarial training and policy gradient. Different from the sequence generation task which only aims to generate sequences similar to the given observations, we leverage adversarial training to help reduce bias in the user model and further reduce the variance in training our agent. The agent learns from both the interactions with the user behavior model and those stored in the logged offline data. To the best of our knowledge, this is the first work that utilizes adversarial training for improving both model approximation and policy learning on offline data.

## 3   Problem Statement

The problem is to learn a policy from offline data such that when deployed online it maximizes cumulative rewards collected from interactions with users. We address this problem with a model-based reinforcement learning solution, which explicitly model users' behavior patterns from data.

**Problem** A recommender is formed as a learning agent to generate actions under a policy, where each action gives a recommendation list of $k$ items. Every time through interactions between the agent and the environment (i.e., users of the system), a set $\Omega$ of $n$ sequences $\Omega = \{\tau_1, ..., \tau_n\}$ is recorded, where $\tau_i$ is the $i$-th sequence containing agent actions, user behaviors and rewards: $\tau_i = \{(a_0^i, c_0^i, r_0^i), (a_1^i, c_1^i, r_1^i), ..., (a_t^i, c_t^i, r_t^i)\}$, $r_t^i$ represents the reward on $a_t^i$ (e.g., make a purchase), and $c_t^i$ is the associated user behavior corresponding to agent's action $a_t^i$ (e.g., click on a recommended item). For simplicity, in the rest of paper, we drop the superscript $i$ to represent a general sequence $\tau$. Based on the observed sequences, a policy $\pi$ is learnt to maximize the expected cumulative reward $\mathbb{E}_{\tau \sim \pi}[\sum_{t=0}^{T} r_t]$, where $T$ is the end time of $\tau$.

**Assumption** To narrow the scope of our discussion, we study a typical type of user behavior, i.e., clicks, and make following assumptions: 1) at each time a user must click on one item from the recommendation list; 2) items not clicked in the recommendation list will not influence the user's future behaviors; 3) rewards only relate to clicked items. For example, when taking the user's purchase as reward, purchases can only happen in the clicked items.

**Learning framework** In a Markov Decision Process, an environment consists of a state set $S$, an action set $A$, a state transition distribution $P : S \times A \times S$, and a reward function $f_r : S \times A \to \mathbb{R}$, which maps a state-action pair to a real-valued scalar. In this paper, the environment is modeled as a user behavior model $\mathcal{U}$, and learnt from offline log data. $S$ is reflected by the interaction history before time $t$, and $P$ captures the transition of user behaviors. In the meanwhile, based on the assumptions mentioned above, at each time $t$, the environment generates user's click $c_t$ on items recommended by an agent $\mathcal{A}$ in $a_t$ based on his/her click probabilities under the current state; and the reward function $f_r$ generates reward $r_t$ for the clicked item $c_t$.

Our recommendation policy is learnt from both offline data and data sampled from the learnt user behavior model, i.e., a model-based RL solution. We incorporate adversarial training in our model-based policy learning to: 1) improve the user model to ensure the sampled data is close to true data distribution; 2) utilize the discriminator to scale rewards from generated sequences to further reduce bias in value estimation. Our proposed solution contains an interactive model constructed by $\mathcal{U}$ and $\mathcal{A}$, and an adversarial policy learning approach. We name the solution as Interactive Recommender GAN, or IRecGAN in short. The overview of our proposed solution is shown in Figure 1.

## 4   Interactive Modeling for Recommendation

We present our interactive model for recommendation, which consists of two components: 1) the user behavior model $\mathcal{U}$ that generates user clicks over the recommended items with corresponding rewards; and 2) the agent $\mathcal{A}$ which generates recommendations according to its policy. $\mathcal{U}$ and $\mathcal{A}$ interact with each other to generate user behavior sequences for adversarial policy learning.

**User behavior model** Given users' click observations $\{c_0, c_1, ..., c_{t-1}\}$, the user behavior model $\mathcal{U}$ first projects the clicked item into an embedding vector $\mathbf{e}^u$ at each time [3]. The state $\mathbf{s}_t^u$ can be represented as a summary of click history, i.e., $\mathbf{s}_t^u = h_u(\mathbf{e}_0^u, \mathbf{e}_1^u, ...\mathbf{e}_{t-1}^u)$. We use a recurrent neural network to model the state transition $P$ on the user side, thus for the state $\mathbf{s}_t^u$ we have,

$$\mathbf{s}_t^u = h^u(\mathbf{s}_{t-1}^u, \mathbf{e}_{t-1}^u),$$

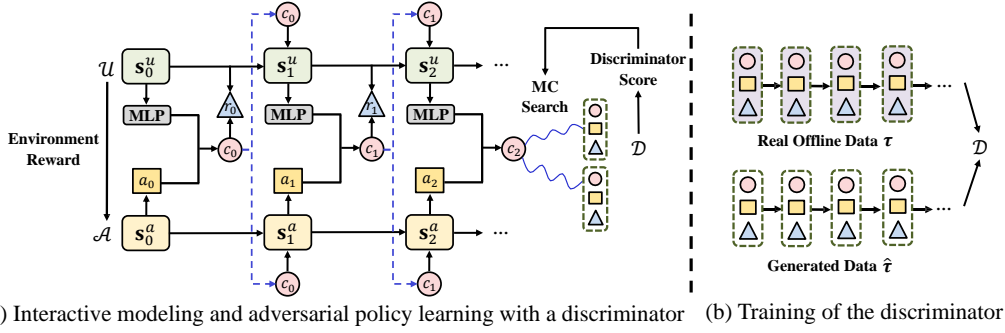

(a) Interactive modeling and adversarial policy learning with a discriminator    (b) Training of the discriminator

Figure 1: Model overview of IRecGAN. $\mathcal{A}, \mathcal{U}$ and $\mathcal{D}$ denote the agent model, user behavior model, and discriminator, respectively. In IRecGAN, $\mathcal{A}$ and $\mathcal{U}$ interact with each other to generate recommendation sequences that are close to the true data distribution, so as to jointly reduce bias in $\mathcal{U}$ and improve the recommendation quality in $\mathcal{A}$.

$h^u(\cdot, \cdot)$ can be functions in the RNN family like GRU [4] and LSTM [12] cells. Given the action $a_t = \{a_{t(1)}, ... a_{t(k)}\}$, i.e., the top-$k$ recommendations at time $t$, we compute the probability of click among the recommended items via a softmax function,

$$\mathbf{V}^c = (\mathbf{W}^c \mathbf{s}_t^u + \mathbf{b}^c)^\top \mathbf{E}_t^u, \;\; p(c_t|\mathbf{s}_t^u, a_t) = \exp(\mathbf{V}_i^c)/\sum\nolimits_{j=1}^{|a_t|} \exp(\mathbf{V}_j^c) \tag{1}$$

where $\mathbf{V}^c \in \mathbb{R}^k$ is a transformed vector indicating the evaluated quality of each recommended item $a_{t(i)}$ under state $\mathbf{s}_t^u$, $\mathbf{E}_t^u$ is the embedding matrix of recommended items, $\mathbf{W}^c$ is the click weight matrix, and $\mathbf{b}^c$ is the corresponding bias term. Under the assumption that target rewards only relate to clicked items, the reward $r_t$ for $(\mathbf{s}_t^u, a_t)$ is calculated by:

$$r_t(\mathbf{s}_t^u, a_t) = f_r\big((\mathbf{W}^r \mathbf{s}_t^u + \mathbf{b}^r)^\top \mathbf{e}_t^u\big), \tag{2}$$

where $\mathbf{W}^r$ is the reward weight matrix, $\mathbf{b}^r$ is the corresponding bias term, and $f_r$ is the reward mapping function and can be set according to the reward definition in specific recommender systems. For example, if we make $r_t$ the purchase of a clicked item $c_t$, where $r_t = 1$ if it is purchased and $r_t = 0$ otherwise, $f_r$ can be realized by a Sigmoid function with binary output.

Based on Eq (1) and (2), taking categorical reward, the user behavior model $\mathcal{U}$ can be estimated from the offline data $\Omega$ via maximum likelihood estimation:

$$\max \sum_{\tau_i \in \Omega} \sum\nolimits_{t=0}^{T_i} \log p(c_t^i|\mathbf{s}_t^{u_i}, a_t^i) + \lambda_p \log p(r_t^i|\mathbf{s}_t^{u_i}, c_t^i), \tag{3}$$

where $\lambda_p$ is a parameter balancing the loss between click prediction and reward prediction, and $T_i$ is the length of the observation sequence $\tau_i$. With a learnt user behavior model, user clicks and reward on the recommendation list can be sampled from Eq (1) and (2) accordingly.

**Agent** The agent should take actions based on the environment's provided states. However, in practice, users' states are not observable in a recommender system. Besides, as discussed in [23], the states for the agent to take actions may be different from those for users to generate clicks and rewards. As a result, we build a different state model on the agent side in $\mathcal{A}$ to learn its states. Similar to that on the user side, given the projected click vectors $\{\mathbf{e}_0^a, \mathbf{e}_2^a, ... \mathbf{e}_{t-1}^a\}$, we model states on the agent side by $\mathbf{s}_t^a = h^a(\mathbf{s}_{t-1}^a, \mathbf{e}_{t-1}^a)$, where $\mathbf{s}_t^a$ denotes the state maintained by the agent at time $t$, $h^a(\cdot, \cdot)$ is the chosen RNN cell. The initial state $\mathbf{s}_0^a$ for the first recommendation is drawn from a distribution $\rho$. We simply denote it as $\mathbf{s}_0$ in the rest of our paper. We should note that although the agent also models states based on users' click history, it might create different state sequences than those on the user side.

Based on the current state $\mathbf{s}_t^a$, the agent generates a size-$k$ recommendation list out of the entire set of items as its action $a_t$. The probability of item $i$ to be included in $a_t$ under the policy $\pi$ is:

$$\pi(i \in a_t|\mathbf{s}_t^a) = \frac{\exp(\mathbf{W}_i^a \mathbf{s}_t^a + \mathbf{b}_i^a)}{\sum_{j=1}^{|C|} \exp(\mathbf{W}_j^a \mathbf{s}_t^a + \mathbf{b}_j^a)}, \tag{4}$$

where $\mathbf{W}_i^a$ is the $i$-th row of the action weight matrix $\mathbf{W}^a$, $C$ is the entire set of recommendation candidates, and $\mathbf{b}_i^a$ is the corresponding bias term. Following [2], we generate $a_t$ by sampling without replacement according to Eq (4). Unlike [3], we do not consider the combinatorial effect among the $k$ items by simply assuming the users will evaluate them independently (as indicated in Eq (1)).

## 5 Adversarial Policy Learning

We use the policy gradient method REINFORCE [34] for the agent's policy learning, based on both the generated and offline data. When generating $\hat{\tau}_{0:t} = \{(\hat{a}_0, \hat{c}_0, \hat{r}_0), ..., (\hat{a}_t, \hat{c}_t, \hat{r}_t)\}$ for $t > 0$, we obtain $\hat{a}_t = \mathcal{A}(\hat{\tau}_{0:t-1}^c)$ by Eq (4), $\hat{c}_t = \mathcal{U}_c(\hat{\tau}_{0:t-1}^c, \hat{a}_t)$ by Eq (2), and $\hat{r}_t = \mathcal{U}_r(\hat{\tau}_{0:t-1}^c, \hat{c}_t)$ by Eq (1). $\tau^c$ represents clicks in the sequence $\tau$ and $(\hat{a}_0, \hat{c}_0, \hat{r}_0)$ is generated by $\mathbf{s}_0^a$ and $\mathbf{s}_0^u$ accordingly. The generation of a sequence ends at the time $t$ if $\hat{c}_t = c_{end}$, where $c_{end}$ is a stopping symbol. The distributions of generated and offline data are denoted as $g$ and $data$ respectively. In the following discussions, we do not explicitly differentiate $\tau$ and $\hat{\tau}$ when the distribution of them is specified. Since we start the training of $\mathcal{U}$ from offline data, it introduces inherent bias from the observations and our specific modeling choices. The bias affects the sequence generation and thus may cause biased value estimation. To reduce the effect of bias, we apply adversarial training to control the training of both $\mathcal{U}$ and $\mathcal{A}$. The discriminator is also used to rescale the generated rewards $\hat{r}$ for policy learning. Therefore, the learning of agent $\mathcal{A}$ considers both sequence generation and target rewards.

### 5.1 Adversarial training

We leverage adversarial training to encourage our IRecGAN model to generate high-quality sequences that capture intrinsic patterns in the real data distribution. A discriminator $\mathcal{D}$ is used to evaluate a given sequence $\tau$, where $\mathcal{D}(\tau)$ represents the probability that $\tau$ is generated from the real recommendation environment. The discriminator can be estimated by minimizing the objective function:

$$-\mathbb{E}_{\tau \sim data} \log\big(\mathcal{D}(\tau)\big) - \mathbb{E}_{\tau \sim g} \log\big(1 - \mathcal{D}(\tau)\big). \tag{5}$$

However, $\mathcal{D}$ only evaluates a completed sequence, and hence it cannot directly evaluate a partially generated sequence at a particular time step $t$. Inspired by [36], we utilize the Monte-Carlo tree search algorithm with the roll-out policy constructed by $\mathcal{U}$ and $\mathcal{A}$ to get sequence generation score at each time. At time $t$, the sequence generation score $q_{\mathcal{D}}$ of $\tau_{0:t}$ is defined as:

$$q_{\mathcal{D}}(\tau_{0:t}) = \begin{cases} \frac{1}{N}\sum_{n=1}^{N} \mathcal{D}(\tau_{0:T}^n), \tau_{0:T}^n \in MC^{\mathcal{U},\mathcal{A}}(\tau_{0:t}; N) & t < T \\ \mathcal{D}(\tau_{0:T}) & t = T \end{cases} \tag{6}$$

where $MC^{\mathcal{U},\mathcal{A}}(\tau_{0:t}; N)$ is the set of $N$ sequences sampled from the interaction between $\mathcal{U}$ and $\mathcal{A}$.

Given the observations in offline data, $\mathcal{U}$ should generate clicks and rewards that reflect intrinsic patterns of the real data distribution. Therefore, $\mathcal{U}$ should maximize the sequence generation objective $\mathbb{E}_{\mathbf{s}_0^u \sim \rho}[\sum_{(a_0,c_0,r_0) \sim g} \mathcal{U}(c_0, r_0|\mathbf{s}_0^u, a_0) \cdot q_{\mathcal{D}}(\tau_{0:0})]$, which is the expected discriminator score for generating a sequence from the initial state. $\mathcal{U}$ may not generate clicks and rewards exactly the same as those in offline data, but the similarity of its generated data to offline data is still an informative signal to evaluate its sequence generation quality. By setting $q_{\mathcal{D}}(\tau_{0:t}) = 1$ at any time $t$ for offline data, we extend this objective to include offline data (it becomes the data likelihood function on offline data). Following [36], based on Eq (1) and Eq (2), the gradient of $\mathcal{U}$'s objective can be derived as,

$$\mathbb{E}_{\tau \sim \{g,data\}}\Big[\sum_{t=0}^{T} q_{\mathcal{D}}(\tau_{0:t})\nabla_{\Theta_u}\big(\log p_{\Theta_u}(c_t|\mathbf{s}_t^u, a_t) + \lambda_p \log p_{\Theta_u}(r_t|\mathbf{s}_t^u, c_t)\big)\Big], \tag{7}$$

where $\Theta_u$ denotes the parameters of $\mathcal{U}$ and $\Theta_a$ denotes those of $\mathcal{A}$. Based on our assumption, even when $\mathcal{U}$ can already capture users' true behavior patterns, it still depends on $\mathcal{A}$ to provide appropriate recommendations to generate clicks and rewards that the discriminator will treat as authentic. Hence, $\mathcal{A}$ and $\mathcal{U}$ are coupled in this adversarial training. To encourage $\mathcal{A}$ to provide *needed* recommendations, we include $q_{\mathcal{D}}(\tau_{0:t})$ as a sequence generation reward for $\mathcal{A}$ at time $t$ as well. As $q_{\mathcal{D}}(\tau_{0:t})$ evaluates the overall generation quality of $\tau_{0:t}$, it ignores sequence generations after $t$. To evaluate the quality of a whole sequence, we require $\mathcal{A}$ to maximize the cumulative sequence generation reward $\mathbb{E}_{\tau \sim \{g,data\}}\big[\sum_{t=0}^{T} q_{\mathcal{D}}(\tau_{0:t})\big]$. Because $\mathcal{A}$ does not directly generate the observations in the interaction sequence, we approximate $\nabla_{\Theta_a}\big(\sum_{t=0}^{T} q_{\mathcal{D}}(\tau_{0:t})\big)$ as 0 when calculating the gradients. Putting these together, the gradient derived from sequence generations for $\mathcal{A}$ is estimated as,

$$\mathbb{E}_{\tau \sim \{g,data\}}\Big[\sum_{t=0}^{T}\big(\sum_{t'=t}^{T}\gamma^{t'-t}q_{\mathcal{D}}(\tau_{0:t})\big)\nabla_{\Theta_a}\log\pi_{\Theta_a}(c_t \in a_t|\mathbf{s}_t^a)\Big]. \tag{8}$$

Based on our assumption that only the clicked items influence user behaviors, and $\mathcal{U}$ only generates rewards on the clicked items, we use $\pi_{\Theta_a}(c_t \in a_t | s_t^a)$ as an estimation of $\pi_{\Theta_a}(a_t | s_t^a)$, i.e., $\mathcal{A}$ should promote $c_t$ in its recommendation at time $t$. In practice, we add a discount factor $\gamma < 1$ when calculating the cumulative rewards to reduce estimation variance [2].

## 5.2 Policy learning

Because our adversarial training encourages IRecGAN to generate clicks and rewards with similar patterns as offline data, and we assume rewards only relate to the clicked items, we use offline data as well as generated data for policy learning. Given data $\tau_{0:T} = \{(a_0, c_0, r_0), ..., (a_T, c_T, r_T)\}$, including both offline and generated data, the objective of the agent is to maximize the expected cumulative reward $\mathbb{E}_{\tau \sim \{g, data\}}[R_T]$, where $R_T = \sum_{t=0}^{T} r_t$. In the generated data, due to the difference in distributions of the generated and offline sequences, the generated reward $\hat{r}_t$ calculated by Eq (2) might be biased. To reduce such bias, we utilize the sequence generation score in Eq (6) to rescale the generated rewards: $r_t^s = q_\mathcal{D}(\tau_{0:t})\hat{r}_t$, and treat it as the reward for generated data. The gradient of the objective is thus estimated by:

$$\mathbb{E}_{\tau \sim \{g, data\}}\Big[\sum_{t=0}^{T} R_t \nabla_{\Theta_a} \log \pi_{\Theta_a}(c_t \in a_t | \mathbf{s}_t^a)\Big], \quad R_t = \sum_{t'=t}^{T} \gamma^{t'-t} q_\mathcal{D}(\tau_{0:t}) r_t \qquad (9)$$

$R_t$ is an approximation of $R_T$ with the discount factor $\gamma$. Overall, the user behavior model $\mathcal{U}$ is updated only by the sequence generation objective defined in Eq (7) on both offline and generated data; but the agent $\mathcal{A}$ is updated by both sequence generation and target rewards. Hence, the overall reward for $\mathcal{A}$ at time $t$ is $q_\mathcal{D}(\tau_{0:t})(1 + \lambda_r r_t)$, where $\lambda_r$ is the weight for cumulative target rewards. The overall gradient for $\mathcal{A}$ is thus:

$$\mathbb{E}_{\tau \sim \{g, data\}}\Big[\sum_{t=0}^{T} R_t^a \nabla_{\Theta_a} \log \pi_{\Theta_a}(c_t \in a_t | \mathbf{s}_t^a)\Big], \quad R_t^a = \sum_{t'=t}^{T} \gamma^{t'-t} q_\mathcal{D}(\tau_{0:t})(1 + \lambda_r r_t) \quad (10)$$

# 6 Theoretical Analysis

For one iteration of policy learning in IRecGAN, we first train the discriminator $\mathcal{D}$ with offline data, which follows $P_{data}$ and was generated by an unknown logging policy, and the data generated by IRecGAN under $\pi_{\Theta_a}$ with the distribution of $g$. When $\Theta_u$ and $\Theta_a$ are learnt, for a given sequence $\tau$, by *proposition* 1 in [9], the optimal discriminator $\mathcal{D}$ is $\mathcal{D}^*(\tau) = \frac{P_{data}(\tau)}{P_{data}(\tau) + P_g(\tau)}$.

**Sequence generation** Both $\mathcal{A}$ and $\mathcal{U}$ contribute to the sequence generation in IRecGAN. $\mathcal{U}$ is updated by the gradient in Eq (7) to maximize the sequence generation objective. At time $t$, the expected sequence generation reward for $\mathcal{A}$ on the generated data is: $E_{\tau_{0:t} \sim g}[q_\mathcal{D}(\tau_{0:t})] = E_{\tau_{0:t} \sim g}[\mathcal{D}(\tau_{0:T} | \tau_{0:t})]$. The expected value on $\tau_{0:t}$ is: $\mathbb{E}_{\tau \sim g}[V_g] = \mathbb{E}_{\tau \sim g}\big[\sum_{t=0}^{T} q_\mathcal{D}(\tau_{0:t})\big] = \sum_{t=0}^{T} \mathbb{E}_{\tau_{0:t} \sim g}\big[\mathcal{D}(\tau_{0:T} | \tau_{0:t})\big]$. Given the optimal $\mathcal{D}^*$, the sequence generation value can be written as:

$$\mathbb{E}_{\tau \sim g}[V_g] = \sum_{t=0}^{T} \mathbb{E}_{\tau_{0:t} \sim g}\Big[\frac{P_{data}(\tau_{0:T} | \tau_{0:t})}{P_{data}(\tau_{0:T} | \tau_{0:t}) + P_g(\tau_{0:T} | \tau_{0:t})}\Big]. \qquad (11)$$

Maximizing each term in the summation of Eq (11) is an objective for the generator at time $t$ in GAN. According to [9], the optimal solution for all such terms is $P_g(\tau_{0:T} | s_0) = P_{data}(\tau_{0:T} | s_0)$. It means $\mathcal{A}$ can maximize the sequence generation value when it helps to generate sequences with the same distribution as $data$. Besides the global optimal, Eq (11) also encourages $\mathcal{A}$ to reward each $P_g(\tau_{0:T} | \tau_{0:t}) = P_{data}(\tau_{0:T} | \tau_{0:t})$, even if $\tau_{0:t}$ is less likely to be generated from $P_g$. This prevents IRecGAN to recommend items only considering users' immediate preferences.

**Value estimation** The agent $\mathcal{A}$ should also be updated to maximize the expected value of target rewards $V_a$. To achieve this, we use discriminator $\mathcal{D}$ to rescale the estimation of $V_a$ on the generated sequences, and we also combine offline data to evaluate $V_a$ for policy $\pi_{\Theta_a}$:

$$\mathbb{E}_{\tau \sim \pi_{\Theta_a}}[V_a] = \lambda_1 \sum_{t=0}^{T} \mathbb{E}_{\tau_{0:t} \sim g} \frac{P_{data}(\tau_{0:t})}{P_{data}(\tau_{0:t}) + P_g(\tau_{0:t})} \hat{r}_t + \lambda_2 \sum_{t=0}^{T} \mathbb{E}_{\tau_{0:t} \sim data} r_t, \qquad (12)$$

where $\hat{r}_t$ is the generated reward by $\mathcal{U}$ at time $t$ and $r_t$ is the true reward. $\lambda_1$ and $\lambda_2$ represent the ratio of generated data and offline data during model training, and we require $\lambda_1 + \lambda_2 = 1$. Here we simplify $P(\tau_{0:T} | \tau_{0:t})$ as $P(\tau_{0:t})$. As a result, there are three sources of biases in this value estimation:

$$\Delta = \hat{r}_t - r_t, \quad \delta_1 = 1 - P_{\pi_{\Theta_a}}(\tau_{0:t})/P_g(\tau_{0:t}), \quad \delta_2 = 1 - P_{\pi_{\Theta_a}}(\tau_{0:t})/P_{data}(\tau_{0:t}).$$

Based on different sources of biases, the expected value estimation in Eq (12) is:

$$\mathbb{E}_{\tau \sim \pi_{\Theta_a}}[V_a] = \lambda_1 \sum_{t=0}^{T} \mathbb{E}_{\tau_{0:t} \sim g} \frac{P_{\pi_{\Theta_a}}(\tau_{0:t})}{P_g(\tau_{0:t})} \frac{\Delta + r_t}{2 - (\delta_1 + \delta_2)} + \lambda_2 \sum_{t=0}^{T} \mathbb{E}_{\tau_{0:t} \sim data} \Big( \frac{P_{\pi_{\Theta_a}}(\tau_{0:t})}{P_{data}(\tau_{0:t})} + \delta_2 \Big) r_t$$

$$= V_a^{\pi_{\Theta_a}} + \sum_{t=0}^{T} \mathbb{E}_{\tau_{0:t} \sim \pi_{\Theta_a}} w_t \Delta + \sum_{t=0}^{T} \mathbb{E}_{\tau_{0:t} \sim data} \lambda_2 \delta_2 r_t - \sum_{t=0}^{T} \mathbb{E}_{\tau_{0:t} \sim \pi_{\Theta_a}} (\lambda_1 - w_t) r_t,$$

where $w_t = \frac{\lambda_1}{2 - (\delta_1 + \delta_2)}$. $\Delta$ and $\delta_1$ come from the bias of user behavior model $\mathcal{U}$. Because the adversarial training helps improve $\mathcal{U}$ to capture real data patterns, it decreases $\Delta$ and $\delta_2$. Because we can adjust the sampling ratio $\lambda_1$ to reduce $w_t$, $w_t \Delta$ can be small. The sequence generation rewards for agent $\mathcal{A}$ encourage distribution $g$ to be close to $data$. Because $\delta_2 = 1 - \frac{P_{\pi_{\Theta_a}}(\tau_{0:t})}{P_g(\tau_{0:t})} \cdot \frac{P_g(\tau_{0:t})}{P_{data}(\tau_{0:t})}$, the bias $\delta_2$ can also be reduced. It shows our method has a bias controlling effect.

# 7 Experiments

In our theoretical analysis, we can find that reducing the model bias improves value estimation, and therefore improves policy learning. In this section, we conduct empirical evaluations on both real-world and synthetic datasets to demonstrate that our solution can effectively model the pattern of data for better recommendations, compared with state-of-the-art solutions.

## 7.1 Simulated Online Test

Subject to the difficulty of deploying a recommender system with real users for online evaluation, we use simulation-based studies to first investigate the effectiveness of our approach following [37, 3].

**Simulated Environment** We synthesize an MDP to simulate an online recommendation environment. It has $m$ states and $n$ items for recommendation, with a randomly initialized transition probability matrix $P(s \in S|a_j \in A, s_i \in S)$. Under each state $s_i$, an item $a_j$'s reward $r(a_j \in A|s_i \in S)$ is uniformly sampled from the range of 0 to 1. During the interaction, given a recommendation list including $k$ items selected from the whole item set by an agent, the simulator first samples an item proportional to its ground-truth reward under the current state $s_i$ as the click candidate. Denote the sampled item as $a_j$, a Bernoulli experiment is performed on this item with $r(a_j)$ as the success probability; then the simulator moves to the next state according to the state transition probability $p(s|a_j, s_i)$. A special state $s_0$ is used to initialize all the sessions, which do not stop until the Bernoulli experiment fails. The immediate reward is 1 if the session continues to the next step; otherwise 0. In our experiment, $m, n$ and $k$ are set to 10, 50 and 10 respectively.

**Offline Data Generation** We generate offline recommendation logs denoted by $d_{\text{off}}$ with the simulator. The bias and variance in $d_{\text{off}}$ are especially controlled by changing the logging policy and the size of $d_{\text{off}}$. We adopt three different logging policies: 1) uniformly random policy $\pi_{\text{random}}$, 2) maximum reward policy $\pi_{\text{max}}$, 3) mixed reward policy $\pi_{\text{mix}}$. Specifically, $\pi_{\text{max}}$ recommends the top $k$ items with the highest ground-truth reward under the current simulator state at each step, while $\pi_{\text{mix}}$ randomly selects $k$ items with either the top 20%-50% ground-truth reward or the highest ground-truth reward under a given state. In the meanwhile, we vary the size of data in $d_{\text{off}}$ from 200 to 10,000.

**Baselines** We compared our IRecGAN with the following baselines: 1) **LSTM:** only the user behavior model trained on offline data; 2) **PG:** only the agent model trained by policy gradient on offline data; 3) **LSTMD:** the user behavior model in IRecGAN, updated by adversarial training.

**Experiment Settings** The hyper-parameters in all models are set as follows: the item embedding dimension is set to 50, the discount factor $\gamma$ in value calculation is set to 0.9, the scale factors $\lambda_r$ and $\lambda_p$ are set to 3 and 1. We use 2-layer LSTM units with 512-dimension hidden states. The ratio of generated training samples and offline data for each training epoch is set to 1:10. We use an RNN based discriminator in all experiments with details provided in the appendix.

**Online Evaluation** After training our models and baselines on $d_{\text{off}}$, we deploy the learned policy to interact with the simulator for online evaluation. We calculated coverage@r to measure the proportion of the true top $r$ relevant items that are ranked in the top $k$ recommended items by a model across all time steps (details in the appendix). The results of coverage@r under different configurations of offline data generation are reported in Figure 2. Under $\pi_{\text{random}}$, coverage@r of all algorithms are relatively low when $r$ is large and the difference in overall performance between behavior and agent

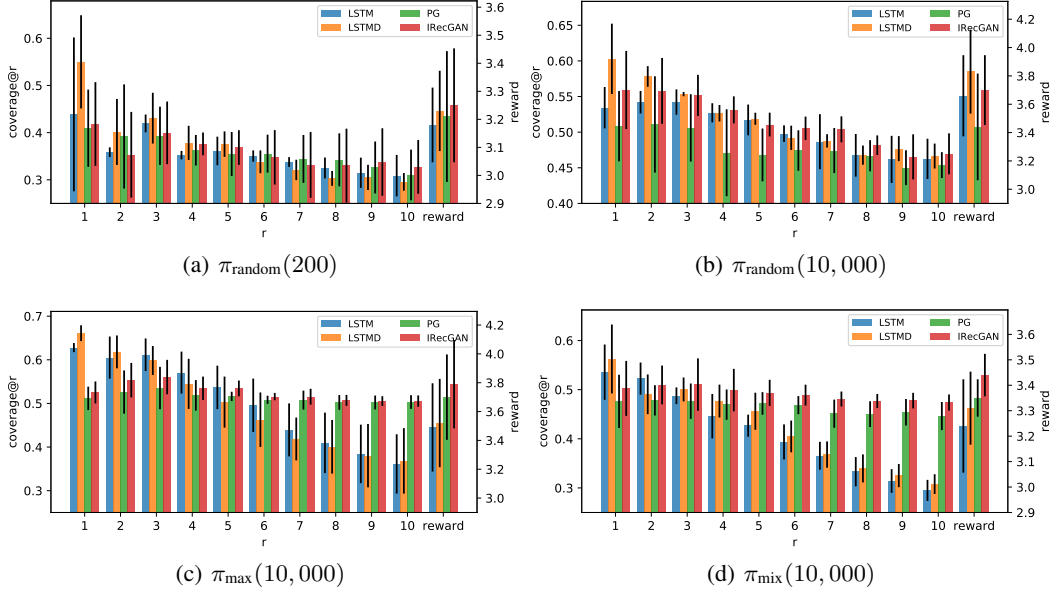

(a) $\pi_{\mathrm{random}}(200)$

(b) $\pi_{\mathrm{random}}(10,000)$

(c) $\pi_{\mathrm{max}}(10,000)$

(d) $\pi_{\mathrm{mix}}(10,000)$

Figure 2: Online evaluation results of coverage@r and cumulative rewards.

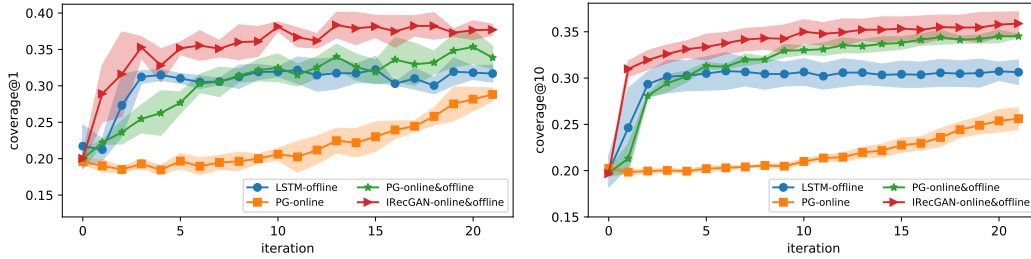

Figure 3: Online learning results of coverage@1 and coverage@10.

models is not very large. This suggests the difficulty of recognizing high reward items under $\pi_{\mathrm{random}}$, because every item has an equal chance to be observed (i.e., full exploration) especially with a small size of offline data. However, under $\pi_{\mathrm{max}}$ and $\pi_{\mathrm{mix}}$, when the high reward items can be sufficiently learned, user behavior models (LSTM, LSTMD) fail to capture the overall preferred items while agent models (PG, IRecGAN) are stable to the change of $r$. IRecGAN shows its advantage especially under $\pi_{\mathrm{mix}}$, which requires a model to differentiate top relevant items from those with moderate reward. It has close coverage@r to LSTM when $r$ is small and better captures users' overall preferences when user behavior models fail seriously. When rewards can not be sufficiently learned (Fig 2(a)), our mechanism can strengthen the influence of truly learned rewards (LSTMD outperforms LSTM when $r$ is small) but may also underestimate some bias. However, when it is feasible to estimate the reward generation (Fig 2(b)(c)(d)), both LSTMD and IRecGAN outperform baselines in coverage@r under the help of generating samples via adversarial training.

The average cumulative rewards are also reported in the rightmost bars of Figure 2. They are calculated by generating 1000 sequences with the environment and take the average of their cumulative rewards. IRecGAN has a larger average cumulative reward than other methods under all configurations except $\pi_{\mathrm{random}}$ with 10,000 offline sequences. Under $\pi_{\mathrm{random}}(10,000)$ IRecGAN outperforms PG but not LSTMD. The low cumulative reward of PG under $\pi_{\mathrm{random}}$ indicates that the transition probabilities conditioned on high rewarded items may not be sufficiently learned under the random offline policy.

**Online Learning** To evaluate our model's effectiveness in a more practical setting, we execute online and offline learning alternately. Specifically, we separate the learning into two stages: first, the agents can directly interact with the simulator to update their policies, and we only allow them to generate 200 sequences in this stage; then they turn to the offline stage to reuse their generated data for offline learning. We iterate the two stages and record their performance in the online learning stage. We compare with the following baselines: 1) **PG-online** with only online learning, 2) **PG-online&offline** with online learning and reusing the generated data via policy gradient for offline

learning, and 3) **LSTM-offline** with only offline learning. We train all the models from scratch and report the performance of coverage@1 and coverage@10 over 20 iterations in Figure 3. We can observe that LSTM-offline performs worse than other RL methods with offline learning, especially in the later stage, due to its lack of exploration. PG-online improves slowly as it does not reuse the generated data. Compared with PG-online&offline, IRecGAN has better convergence and coverage because of its reduced value estimation bias. We also find that coverage@10 is harder to improve. The key reason is that as the model identifies the items with high rewards, it tends to recommend them more often. This gives less relevant items less chance to be explored, which is similar to our online evaluation experiments under $\pi_{\max}$ and $\pi_{\min}$. Our model-based RL training alleviates this bias to a certain extent by generating more training sequences, but it cannot totally alleviate it. This reminds us to focus on explore-exploit trade-off in model-based RL in our future work.

## 7.2 Real-world Data Offline Test

We use a large-scale real-world recommendation dataset from CIKM Cup 2016 to evaluate the effectiveness of our proposed solution for offline reranking. Sessions of length 1 or longer than 40 and items that have never been clicked are filtered out. We selected the top 40,000 most popular items into the recommendation candidate set, and randomly selected 65,284/1,718/1,720 sessions for training/validation/testing. The average length of sessions is 2.81/2.80/2.77 respectively; and the ratio of clicks which lead to purchases is 2.31%/2.46%/2.45%. We followed the same model setting as in our simulation-based study in this experiment. To understand of the effect of different data separation strategies on RL model training and test, we also provide a comparison of performances under different data separation strategies in the appendix.

**Baselines** In addition to the baselines we compared in our simulation-based study, we also include the following state-of-the-art solutions for recommendation: 1). **PGIS:** the agent model estimated with importance sampling on offline data to reduce bias; 2). **AC:** an LSTM model whose setting is the same as our agent model but trained with actor-critic algorithm [16] to reduce variance; 3). **PGU:** the agent model trained using offline and generated data, without adversarial training; 4). **ACU:** AC model trained with both offline and generated data, without adversarial training.

**Evaluation Metrics** All the models were applied to rerank the given recommendation list at each step of testing sessions in offline data. We used Precision@k (P@1 and P@10) to compare different models' recommendation performance, where we define the clicked items as relevant. Because the logged recommendation list was not ordered, we cannot assess the logging policy's performance here.

Table 1: Rerank evaluation on real-world dataset with random splitting.

| Model | LSTM | LSTMD | PG | PGIS | AC | PGU | ACU | IRecGAN |
|---|---|---|---|---|---|---|---|---|
| P@10 (%) | 32.89±0.50 | 33.42±0.40 | 33.28±0.71 | 28.13±0.45 | 31.93±0.17 | 34.12±0.52 | 32.43±0.22 | **35.06**±0.48 |
| P@1 (%) | 8.20±0.65 | **8.55**±0.63 | 6.25±0.14 | 4.61±0.73 | 6.54±0.19 | 6.44±0.56 | 6.63± 0.29 | 6.79±0.44 |

**Results** The results of the offline rerank evaluation are reported in Table 1. With the help of adversarial training, IRecGAN achieved encouraging P@10 improvement against all baselines. This verifies the effectiveness of our model-based reinforcement learning, especially its adversarial training strategy for utilizing the offline data with reduced bias. Specifically, PGIS did not perform as well as PG partially because of the high variance introduced by importance sampling. PGU was able to fit the given data more accurately than PG by learning from the generated data, since there are many items for recommendation and the collected data is limited. However, PGU performed worse than IRecGAN because of the biased user behavior model. And with the help of the discriminator, IRecGAN reduces the bias in the user behavior model to improve value estimation and policy learning. This is also reflected on its improved user behavior model: LSTMD outperformed LSTM, given both of them are for user behavior modeling.

## 8 Conclusion

In this work, we developed a practical solution for utilizing offline data to build a model-based reinforcement learning solution for recommendation. We introduce adversarial training for joint user behavior model learning and policy update. Our theoretical analysis shows our solution's promise in reducing bias; our empirical evaluations in both synthetic and real-world recommendation datasets verify the effectiveness of our solution. Several directions left open in our work, including balancing explore-exploit in policy learning with offline data, incorporating richer structures in user behavior modeling, and exploring the applicability of our solution in other off-policy learning scenarios, such as conversational systems.

## Footnotes

[2]Our implementation is available at `https://github.com/JianGuanTHU/IRecGAN`.

[3]As we can use different embeddings on the user side and agent side, we use the superscript $u$ and $a$ to denote this difference accordingly.

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
