[Supplementary Material]

# Appendix

## 1 Details of Discriminator Model

We adopt an RNN-based discriminator $\mathcal{D}$ for our IRecGAN framework, and model its hidden states by $\mathbf{s}_t^d = h_d(\mathbf{s}_{t-1}^d, \mathbf{e}_{t-1}^d)$, where $\mathbf{s}_t^d$ denotes the hidden states maintained by the discriminator at time $t$ and $\mathbf{e}_{t-1}^d$ is the embedding used in the discriminator side. And we add a multi-layer perceptron which takes the hidden states as input to compute a score through a Sigmoid layer indicating whether the trajectory is likely to be generated by real users when interacting with a recommender as following:

$$D(\tau_{0:T}) = \text{Sigmoid}\big[\frac{1}{T}\sum_{t=0}^{T}\mathbf{e}^d(c_{a_t}^{\max})^\top \mathbf{e}^d(c_t)(\mathbf{W}^p r_t + \mathbf{b}^p)\big]$$

$$c_{a_t}^{\max} = \text{argmax}_{c\in a_t}(\mathbf{W}^d\mathbf{s}_t^d + \mathbf{b}^d)^\top \mathbf{e}(c)$$

where $c_{a_t}^{\max}$ can be considered as the user's preferred item in the given recommendation list $a_t$, and should be as close to the observed clicks $c_t$ as possible for real users. To enable the gradient backpropagation, we use Softmax with a temperature 0.1 to approximate the argmax function. Other hyper parameters are set to the same with the experiment setting depicted in Section 7.1. The optimization target of $\mathcal{D}$ is formulated as in Eq (5).

## 2 Details of Sampling

To enable exploration during model training, inspired by the discussion in [2], we sample items to get the recommendation list (action) by Eq (4). Moreover, since our model-based RL solution involves the user behavior model (which is estimated together with the agent model) during the sequence generation, we sample users' clicks by their probabilities in the sequence generation of training as well. For testing, the agent's recommendation list contains items with top $k$ probabilities under the learned policy. In the meanwhile, for comparison purpose, we can also use the user behavior model to create a ranking list of items for recommendation purpose. Specifically, in the offline evaluation, user behavior models rerank recorded offline recommendations; and in the simulated online evaluation, user behavior models rerank all items in $A$ (i.e., with given recommendations containing all items) and select the top $k$ for the evaluation of coverage@r.

## 3 Algorithm

---
**Algorithm 1:** IRecGAN
---
**Input:** Offline data; an agent model $\mathcal{A}$; a user behavior model $\mathcal{U}$; a discriminator $\mathcal{D}$.

1  Initialize an empty simulated sequences set $B^s$ and a real sequences set $B^r$.
2  Initialize $\mathcal{U}, \mathcal{A}$ and $\mathcal{D}$ with random parameters.
3  Pre-train $\mathcal{U}$ by maximizing Eq (3).
4  Pre-train $\mathcal{A}$ via the policy gradient of Eq (10) using only the offline data.
5  $\mathcal{A}$ and $\mathcal{U}$ simulate $m$ sequences and add them to $B^s$. Add $m$ trajectories to real data set $B^r$.
6  Pre-train $\mathcal{D}$ according to Eq (5) using $B^r$ and $B^s$.
7  **for** $e \leftarrow 1$ **to** $epoch$ **do**
8      **for** r − steps **do**
9          Empty $B^s$ and then generate $m$ simulated sequences and add to $B^s$.
10         Compute $q_{\mathcal{D}}(\tau_{0:t})$ at each step $t$ by Eq (6).
11         Extract $\left\lfloor\frac{\lambda_2}{\lambda_1}m\right\rfloor$ sequences into $B^r$.
12         Update $\mathcal{U}$ via the policy gradient of Eq (7) with $B = [B^s, B^r]$.
13         Update $\mathcal{A}$ via the policy gradient of Eq (10) with $B = [B^s, B^r]$.
14     **end**
15     **for** d − steps **do**
16         Empty $B^s$, then generate $m$ simulated sequences by current $\mathcal{U}, \mathcal{A}$ and add to $B^s$.
17         Empty $B^r$ and add $m$ sequences from the offline data.
18         Update $\mathcal{D}$ according to Eq (5) for $i$ epochs using $B^r$ and $B^s$.
19     **end**
20 **end**
---

# 4 The Weight of Sequence Generation Score

A weight $w$ can be applied to the sequence generation score $q_D$ for purpose of rescaling the generated rewards. In this paper, we set $w = 1$ and got the expected value estimation in Section 6. In this setting, when the agent's policy is the same as that of the offline data and the user behavior model is unbiased, which means $P_{\pi_{\Theta_a}}(\tau_{0:t}) = P_g(\tau_{0:t}) = P_{data}(\tau_{0:t})$ and $\Delta = 0$, the value estimation is biased. By setting $w = 2$, the expected value estimation $\mathbb{E}_{\tau \sim \pi_{\Theta_a}}[V_a]$ turns out to be:

$$V_a^{\pi_{\Theta_a}} + \sum_{t=0}^{T} \mathbb{E}_{\tau_{0:t} \sim \pi_{\Theta_a}} \frac{2\lambda_1}{2 - (\delta_1 + \delta_2)}\Delta + \sum_{t=0}^{T} \mathbb{E}_{\tau_{0:t} \sim data}\lambda_2 \delta_2 r_t(\tau_{0:t}) + \sum_{t=0}^{T} \mathbb{E}_{\tau_{0:t} \sim \pi_{\Theta_a}} \frac{(\delta_1 + \delta_2)\lambda_1}{2 - (\delta_1 + \delta_2)} r_t(\tau_{0:t}).$$

This value estimation is unbiased when $P_{\pi_{\Theta_a}}(\tau_{0:t}) = P_g(\tau_{0:t}) = P_{data}(\tau_{0:t})$ and $\Delta = 0$.

However, when the user behavior model is biased, amplifying the sequence generation score with $w > 1$ will also amplify the bias. Moreover, it will over-penalize the generated sequences which are not very similar to the offline data (with relatively low $q_D$). Although our method encourages the agent to consider users' immediate clicks when making recommendations, it does not require the overall recommendations to be similar to those of the offline data. And we also do not want the agent's recommendations to be exactly the same as the recorded ones, since our goal is to improve the offline policy. In this case, over-penalizing some generated sequences is harmful. Because of the reasons above, we directly use $q_D$ in our paper. But we admit that the weight $w$ can be set to different values under specific cases for value estimation.

# 5 Details about the Coverage Metric

In our simulated environment, the selection of a click directly relates to its reward, which also influences the length of the sequence. In this case, whether the model (the user behavior model or the agent) can capture real reward of items at each time will highly affect its performance in both behavior prediction and recommendation. As indicated in Section 7.1, we use coverage@r to measure whether a model can capture items with high rewards (most relevant items) under corresponding states. Denote the top $r$ relevant items at time $t$ as $C_t^r$, the top $k$ recommendations given by the model as $A_t^k$. We regard the $k$ items with the highest prediction scores from a recommendation algorithm (a user behavior model can also be treated as a recommendation algorithm when the click candidates are from the whole item set) as its recommendations given the whole item set as its candidates. Then the coverage@r can be calculated by

$$\text{coverage@r} = \frac{\sum_{t=0}^{T} \left| C_t^r \cap A_t^k \right|}{T \times r}.$$

When $r$ is small, it requires models to capture the most relevant items to get a high coverage@r. When $r$ becomes larger, models which can capture overall high reward items are likely to get high coverage@r. For example, an evaluation result with high coverage@1 and low coverage@2 indicates the algorithm handles the highest reward item in the ground-truth better than the second item.

To the behavior model, since the environment's next clicks are sampled according to the items' conditional rewards with respect to the state, a model's coverage@r performance directly relates to its performance of the behavior prediction (especially when $r$ is small). To the agent, since it aims to maximize the cumulative rewards, including items with relatively high immediate rewards will ensure users' satisfaction and the model's immediate gain at each time step. Moreover, since items with high rewards also have high success probabilities in the Bernoulli experiment, ensuring users' clicking of high reward items encourages the continuation of sequences, which also improves the accumulation of rewards. Because of these, the agent's coverage@r performance is highly related to the actual cumulative rewards it can get in our simulated environment. However, different from the behavior model, because the cumulative rewards an agent can get also relate to the state transitions conditioned on the clicked items, a performing agent should not always recommend items with the highest rewards. In this case, we also provide an evaluation of cumulative rewards in the results.

# 6 Correction of Figure 2 and Figure 3

Compared to the original version, we have corrected Figure 2 and Figure 3 about the results of simulated experiments. This is because of an implementation mismatch when computing the simulated environment. Specifically, in the previous implementation the next click under the state $s_i$ is re-selected by $\arg\max_{a_i} r(a_i \in A_t^k | s_i)$ instead of $a_j$, after the success of the Bernoulli experiment with the probability $r(a_j | s_i)$. This leads to a situation in which all methods are hard to estimate rewards

of items with relatively low ground-truth rewards in $s_i$, no matter under $\pi_{\text{random}}$ or $\pi_{\text{max}}$. This leads to the performance drop of coverage@r with the increase of $r$. After the correction, the updated results and their corresponding analysis are shown in Section 7.1.

# 7 Offline Test with Different Data Separations

In real-world data offline evaluation of Section 7.2, since we do not know the logging policy of the offline recommendation, the true distribution of data appearing under the offline recommendation policy can only be inferred by the observations. However, because the problem space of our offline dataset is large, it is hard to sufficiently reveal the true data distribution with limited offline data. In this case, using different data separation strategies may lead to different data distributions for both training and testing, which may cause different performance of models as indicated in our simulated online evaluation. To provide a more comprehensive evaluation, we randomly split the dataset for training/validation/testing in Section 7.2. We adopted P@1 and P@10 to compare different models' performance. And both the metrics were calculated only on the timesteps with a recommendation list including more than 10 candidate items. Moreover, we conducted the offline evaluation experiments three times by varying the random seed to get the confidence interval for each algorithm.

To compare results under different data separation strategies, we evaluated models when splitting the dataset in the order of session ID or time, as shown in Table 2 and 3, respectively. Specifically, we ordered the whole dataset by session ID or time, and used 65,284/1,718/1,820 sessions for training/validation/testing. When split data by session ID, the average length of training/validation/testing sessions was 2.84/2.15/2.09, the ratio of clicks that lead to purchases was 2.32%/2.08%/2.36%. When split data by time, the average length of training/validation/testing sessions was 2.81/2.80/2.75 and the ratio of clicks leading to purchases was 2.33%/2.21%/2.05%. And to provide more insights about performance of different algorithms, we also included P@1 (all) that measures Precision@1 on all the timesteps (with more than one recommendation candidate) for each model as a metric.

Table 2: Rerank evaluation on real-world recommendation dataset when split by session ID.

| Model | LSTM | LSTMD | PG | PGIS | AC | PGU | ACU | IRecGAN |
|---|---|---|---|---|---|---|---|---|
| P@10 (%) | 28.79±0.44 | 31.98±0.64 | 32.44±1.16 | 30.72±0.37 | 29.26±0.79 | 30.33±0.47 | 28.53±0.35 | **33.45**±0.71 |
| P@1 (%, all) | 9.64±0.38 | **11.26**±0.34 | 8.40±0.18 | 7.67±0.31 | 7.33±0.41 | 8.27±0.44 | 7.08±0.32 | 9.78±0.37 |
| P@1 (%) | 9.68±0.29 | **11.06**±0.23 | 6.83±0.38 | 6.09±0.19 | 6.11±0.18 | 6.67±0.51 | 5.86±0.26 | 7.84±0.25 |

Table 3: Rerank evaluation on real-world recommendation dataset when split by time.

| Model | LSTM | LSTMD | PG | PGIS | AC | PGU | ACU | IRecGAN |
|---|---|---|---|---|---|---|---|---|
| P@10 (%) | 27.95±0.34 | 29.85±0.18 | 29.13±0.18 | 27.85±0.15 | 25.37±0.49 | 29.45±0.37 | 26.51±0.67 | **30.07**±0.15 |
| P@1 (%, all) | 7.94±0.10 | **8.27**±0.14 | 6.07±0.15 | 6.91±0.11 | 4.08±0.12 | 6.58±0.18 | 4.84±0.23 | 7.08±0.25 |
| P@1 (%) | 7.67±0.12 | **7.90**±0.14 | 4.65±0.25 | 5.40±0.13 | 4.16±0.15 | 5.19±0.27 | 4.89±0.21 | 5.81±0.18 |

We observed that the results had a considerable difference compared with random data separation when we split the data by session ID or time, which validated the influence of data separation. However, the overall conclusions in the comparison among our methods (LSTMD, IRecGAN) and baselines remained consistent. Because of their different training purposes where user behavior models (LSTM, LSTMD) were trained only for click prediction, LSTM and LSTMD performed better than the RL agents in P@1. And the RL agents (IRecGAN and other RL baselines) had advantages in capturing users' overall interests, which led to better P@10 results.

Although we observed different performance of baselines under different data separation strategies, using our additional sample generation mechanism with adversarial training, under all strategies LSTMD outperformed LSTM, IRecGAN outperformed other RL-based methods in both P@1 and P@10. These results showed that 1) the solution we proposed in this paper helped to improve the user behavior model. 2) The sequence generation reward for the agent helped it better capture users' immediate behaviors. 3) The proposed solution helped the agent to better capture users' overall interests.

By comparing the P@1 and P@1 (all), we observed that the differences between these two metrics in user behavior models (LSTM and LSTMD) were small while those of most RL agents were relatively large. More specifically, most of RL agents performed better under the P@1 (all) metric than P@1, where the former included evaluations with less than 10 ranking candidates. We conjecture that the key reason is user behavior models are only optimized for click prediction, while agents need to balance both the next click and future clicks via the learnt state transition. When the number of recommendation candidates to re-rank is small, there is more chance that an agent ranks the next click to be the first, which leads to a better P@1 (all).