[Reviews · NeurIPS 2019]

Reviewer 1



Summary: This work proposed a practical implementation of an effective approach InteractiveRecGan to utilize offline data to build a recommender agents with better policy than the reference one. Specifically, a model-based RL framework is incorporated with adversarial training to better utilize the offline logged data to lean a “optimal” policy. Using 1 - 5 scale with 5 highest Originality: 3 out of 5 Compared with the recent related works such as [1], the main difference is to use a model-based RL instead of a model-free RL. Meanwhile, compared with model-based RL in recommendation such as [2], the main difference is the adversarial training Quality: 4 out of 5 This work is technically sound, and the proposed methods are well supported by both theoretical analysis and the empirical analysis Clarity: 4 out of 5 This paper is very well-written and the empirical results are presented in an easy-to-read way. All the plots are very easy to read Significance: 3 to 4 out of 5 It is not clear and how promising the proposed approach is to be deployed and served as an online recommender [1]: Generative Adversarial User Model for Reinforcement Learning Based Recommendation System, in ICML'19 [2]: Model-Based Reinforcement Learning for Whole-Chain Recommendations

Reviewer 2



In this paper, the authors propose InteractiveRecGan to utilize offline data and build a recommender agents with model-based RL algorithms. They integrate a user behavior model, an agent model and a discriminator together. The paper is well written and easy to follow. The proposed model is verified by theoretical analysis and experiments with state-of-art baselines. I have read authors' response, but my mind did not change.

Reviewer 3



Originality: The proposed approach is a novel combination of well-known techniques such as RL and GAN for recommendation. Related work has been adequately cited. It is clear how the proposed approach differs from the existing literature. Quality: The approach appears to be technically sound. The theoretical analysis and the experiments support the claims. This is a complete piece of work. The experiments and results sections are quite terse. I wonder if authors could have identified any more experiments relevant to their work and cut down some of the theory/derivation (or, move them to the supplementary section) to make space. line 271: We randomly selected 65,284 sessions for training and the left 3,437 for testing. Were the parameters tuned on the test set? Did you have a Validation set? Clarity: The paper is mostly well-written except for some typos: line 32: we  **exploring framing** recommendation as building reinforcement learning (RL) agents line 37: Classic model-free RL applications requiring collecting large quantities of interaction data with self-play and simulation. line 39: In contrast, such methods suffer from very high sample complexity so that simulation for generating realistic interaction **experience nonviable**. line 169: And at each step, **the will** generate next click by line 174: Simultaneously the model **needs to decides** the probability line 248: The last term is the **objection** for generator in GAN. Significance: The overall problem of learning to recommend is an important problem. Any approach that can improve recommendation performance will be of interest to multiple entities - industry, academia. Statistical significance numbers are missing in the results. Otherwise, the results look good. I have carefully considered the authors' response. The rebuttal looks fair. However, my questions were more of a request for clarification. So, it doesn't change my score.

[Author Response · NeurIPS 2019]

First, we sincerely thank all reviewers for their thoughtful comments and suggestions. In the following, we will first respond to the common suggestions from all three reviewers, and then answer specific questions from each reviewer.

**Balance between theoretical analysis and empirical evaluation:** The key motivation of this work is to investigate the value of generative adversarial training for model-based reinforcement learning (RL) with offline data, especially how the discriminator could help control bias in the estimated model. Hence, we put an emphasis on the theoretical analysis when organizing this paper. But we also agree that some intermediate theoretical derivations can be moved to the supplementary file to save more space for experiment results and analysis. We will reorganize this paper with increased content on empirical evaluations (e.g., adding a simulation-based study) in our revision.

**More empirical evaluations:** Our theoretical analysis and reported offline experiments confirm that the proposed approach controls bias in model-based RL and leads to better recommendation quality. But we acknowledge that more datasets/settings can give a more comprehensive understanding of our approach. Subject to the complexity and difficulty of deploying a recommender system with real users for online evaluation, we plan to include simulation-based studies to further investigate the effectiveness of our approach, if this paper got accepted. For example, following [1,2], we can simulate a recommendation environment based on observed user click data; and in this case, we can definitely simulate the environment based on more datasets.

**Statistical significance:** We will report the variance and statistical significance of our empirical results in our revision.

**Responses to Reviewer 1:**

**Serve as an online recommender**: Our theoretical analysis shows that the proposed approach controls bias in model-based RL, and our empirical results on a large-scale offline recommendation dataset confirm the effectiveness of our approach. These shed light on the approach's effectiveness as an online recommender. We also believe that, arguably, a more direct proof of an RL algorithm's effectiveness is the real-world deployment, despite its difficulty in getting the real user population. We plan to follow [1,2] to further evaluate our solution and its potential for future real-world online deployment.

**Challenge in combining the model-based RL and adversarial training:** The key challenge is how to design the discriminator for bias control. In our approach, the discriminator is utilized for two purposes: a) generating sequences similar to offline data, and b) reweighting rewards for value estimation. Without reweighting, higher bias could be introduced in the value estimation. And regularizing the generation of similar sequences ensures $p_g \approx p_{data}$ ($p_g$ is the simplified notation of $P_g(\tau)$, where $\tau$ is the sequence), and thus $\delta_2 = 1 - (p_{\pi_{\theta_a}}/p_g)(p_g/p_{data}) \approx 1 - p_{\pi_{\theta_a}}/p_g$. As a result, if the learnt interactive model is effective ($p_{\pi_{\theta_a}} \approx p_g$), $\delta_2$ would be small. Smaller $\delta_2$ leads to less biased value estimation. These two factors help control bias in value estimation for model-based RL.

**Responses to Reviewer 2:**

Thanks for your helpful suggestions. Please refer to Line 9-15 for our responses to possible new empirical evaluations.

**Responses to Reviewer 3:**

**Dataset split:** Sorry for the confusing description of our dataset split, and we will definitely clarify this in our revision. Our notion of the training dataset (65,284 sessions) actually included both training data (61,847 sessions) and validation data (3,437 sessions, the same size as the testing set). All models' parameters (ours and baselines') were tuned on the validation set. The performance on the testing set was reported in our paper.

**Case study:** Our PGUD recommends items based on both long-term rewards and users' behaviors (clicks). This may cost its worse performance on some sequences which have high repetitions, since most repetitions are associated with zero rewards in our offline data. Here we list one case study as our error analysis. For a session with clicks {29261, 29261, 29261} and rewards {1, 1, 1}, LSTM recommended item 29261 (ranked within top-10) after the second click while PGUD did not at all. This is because LSTM ranks items only by user clicks and it favors repetitions. But in our offline data, repeated items are mostly linked with zero rewards; and thus PGUD did not promote item 29261, as its predicted value is low. However, as we reported in the experiment results, PGUD had a better overall recommendation quality: more than 70% time PGUD ranked rewarded items higher than LSTM, and 55% time PGUD ranked rewarded items higher than policy gradient(PG).

**Limitations:** As a model-based RL solution, bias is always its intrinsic limitation. Since our focus is on offline model estimation, the issue becomes even more serious. Our work utilizes adversarial training to control bias, but cannot completely remove it. As our future work, it is necessary to incorporate other off-policy evaluation methods, such as inverse propensity reweighting and doubly robust estimation, to better balance the bias and variance trade-off in offline RL. It is also necessary to combine our offline learning with online RL methods to best utilize the available data.

**Reference**

[1] Zhao, X., Xia, L., Zhao, Y., Yin, D., and Tang, J. (2019). Model-Based Reinforcement Learning for Whole-Chain Recommendations. arXiv preprint arXiv:1902.03987.

[2] Chen, X., Li, S., Li, H., Jiang, S., Qi, Y., and Song, L. (2019, May). Generative Adversarial User Model for Reinforcement Learning Based Recommendation System. In ICML (pp. 1052-1061).


[Meta-Review · NeurIPS 2019]

The reviewers overall felt positively about this paper, though scores were somewhat marginal. The reviews, while not glowing, overall lean toward acceptance of the paper: the reviewers feel the work is technically sound, the method is practical and effective, related work is good, and the experiments are convincing. There are some more tentative comments regarding the novelty/originality, being mostly a combination of existing techniques (R3), however this issue seems not to be a dealbreaker, and the difference compared to existing work is clear. There are a few clarifying points that seem to be addressed in the rebuttal.